# LncRNA SNHG1 Facilitates Tumor Proliferation and Represses Apoptosis by Regulating PPARγ Ubiquitination in Bladder Cancer

**DOI:** 10.3390/cancers14194740

**Published:** 2022-09-28

**Authors:** Hongzhou Cai, Haifei Xu, Hongcheng Lu, Weizhang Xu, Haofeng Liu, Xinwei Wang, Guoren Zhou, Xuejian Yang

**Affiliations:** 1Department of Urology, Jiangsu Cancer Hospital & The Affiliated Cancer Hospital of Nanjing Medical University & Jiangsu Institute of Cancer Research, Nanjing 210009, China; 2Department of Urology, Nantong Tumor Hospital, Nantong 226361, China; 3Department of Urology, Zhongda Hospital Affiliated to Southeastern China University, Nanjing 210029, China; 4Department of General Surgery, Nantong Tumor Hospital, Nantong 226361, China; 5Department of Oncology, Jiangsu Cancer Hospital & The Affiliated Cancer Hospital of Nanjing Medical University & Jiangsu Institute of Cancer Research, Nanjing 210009, China; 6Department of Urology, Suqian First Hospital, Suqian 223800, China

**Keywords:** bladder cancer, SNHG1, microRNA-9-3p, MDM2, PPARγ

## Abstract

**Simple Summary:**

Our study elucidated that SNHG1 promotes MDM2 expression by binding to miR-9-3p to promote PPARγ ubiquitination and downregulate PPARγ expression and that SNHG1 plays an important role in bladder cancer and provides a potential therapeutic target for bladder cancer.

**Abstract:**

Background: Long noncoding RNAs regulate various biological effects in the progression of cancers. We found that the expression of SNHG1 was significantly up-regulated in bladder cancer after analyzing data obtained from TCGA and GEO. However, the potential role of SNHG1 remains to be investigated in bladder cancer. It was validated that SNHG1 was overexpressed in bladder cancer tissues detected by qRT-PCR and FISH, which was also associated with poor clinical outcome. Additionally, SNHG1 was verified to facilitate tumor proliferation and repress apoptosis in vitro and in vivo. Results: SNHG1 could act as a competitive endogenous RNA and decrease the expression of murine double minute 2 (MDM2) by sponging microRNA-9-3p. Furthermore, MDM2 induced ubiquitination and degradation of PPARγ that contributed to the development of bladder cancer. Conclusions: the study elucidated that SNHG1 played an important role in bladder cancer and provided a potential therapeutic target for bladder cancer.

## 1. Background

Bladder cancer (BCa) ranks 12th in cancer incidence globally, with nearly 570,000 new cases each year and 13th in terms of deaths [1]. A strong male predominance is observed in bladder cancer, where three-fourths of cases occur in men [2]. The increased risk for bladder cancer correlates to factors including age, smoking, exposure to some industrial chemicals and hormonal differences, particularly androgens [3]. The treatment of bladder cancer depends on stage and grade to a great extent, which are also strongly associated with the prognosis of patients: the treatment of non-muscle invasive bladder cancer is usually through resection and immunotherapy with intravesical drugs such as Bacillus Calmette-Guerin, whilst more aggressive methods, such as radical cystectomy coupled with chemotherapy, are necessary for muscle invasive bladder cancer [4]. Unfortunately, little improvement has been achieved in the treatment for bladder cancer with a flat 5-year survival rate until recently [5]. Therefore, it is urgent to get a deeper understanding of molecular mechanisms underlying bladder cancer, thus exploring novel targets for bladder cancer treatment.

It is well-established that long noncoding RNA small nucleolar RNA host gene 1 (lncRNA SNHG1) is involved in tumor stage, size and overall survival [6]. The oncogenic role of SNHG1 has been elucidated in various cancers. For instance, a prior study also reported the tumor-promoting potential of SNHG1 in pancreatic cancer with the results that SNHG1 silencing triggered repression of cell proliferative, metastatic, and invasive capacities by inactivating Notch-1 pathway [7]. In addition, Li et al. observed the suppressive effect of SNHG1 on prostate cancer development by promoting cell proliferation [8]. These findings indirectly support the possibility that SNHG1 might promote development of bladder cancer. Moreover, the starBase website used in our study predicted a binding relationship between SNHG1 and microRNA-9-3p (miR-9-3p).

MiR-9-3p (previously known as miR-9) is widely known for its altered expression and function in multiple diseases, such as Huntington’s disease and cancers [9]. Interestingly, it was detected that ectopically expressed miR-9-3p possessed antitumor potential in bladder cancer by diminishing cell invasion, migration, and proliferation [10]. In our study, the binding sites between miR-9-3p and 3′ untranslated region (UTR) of murine double minute 2 (MDM2) were predicted by TargetScan. Overexpression of MDM2 could reportedly neutralize the depressive effect of miR-379-5p on bladder cancer cell proliferative, migratory and invasive capacities [11]. In the presence of EGFR, MDM2 can bind to peroxisome proliferator-activated receptor-gamma (PPARγ) and regulate the ubiquitination of PPARγ protein in colon cancer cells [12]. More importantly, it was elaborated in a prior study that an antagonist of PPARγ promoted cell cycle entry and decreased cell apoptosis in bladder cancer [13].

Taken these findings into account, we hypothesized that the SNHG1/miR-9-3p/MDM2/PPARγ axis correlated to the progression of bladder cancer. Therefore, the present study was implemented by focusing on the alteration in SNHG1 expression and function in bladder cancer. Moreover, we investigated the underlying mechanism of SNHG1 in the bladder cancer process via miR-9-3p/MDM2/PPARγ axis.

## 2. Methods

### 2.1. Compliance with Ethical Standards

The experiments involving 67 human being samples were approved with ratification of the Ethics Committee of Nanjing Medical University (NO. 402, 2021) by conforming to the principles outlined in the *Declaration of Helsinki*. Ethical agreements were obtained from the donors or their families through written informed consent. Animal experiments (20 healthy nude mice aged 6–8 weeks) were ratified by the Animal Ethics Committee of Nanjing Medical University (No. 402, 2021) and concurred with the Guidelines for Animal Experiments of Peking University Health Science Center.

### 2.2. Patients and Tissue Samples

Sixty-seven patients diagnosed with bladder cancer undergoing radical cystectomy in Suqian First Hospital and Jiangsu Cancer Hospital from June 2016 to June 2017 were enrolled. Fresh bladder cancer tissues and corresponding adjacent normal tissues were preserved in liquid nitrogen immediately after resection. None of patients received preoperative radiotherapy, chemotherapy or immunotherapy. Follow-up information was obtained from outpatient clinics and regular telephone interviews.

### 2.3. Bioinformatics Methods

Gene Expression Profiling Interactive Analysis (GEPIA) was adopted to analyze the BLCA dataset of The Cancer Genome Atlas (TCGA) database to obtain the genes with significant differences (*p* < 0.05), from which the genes with |logFC| > 0.5 were screened out. The Gene Expression Omnibus (GEO) database: https://www.ncbi.nlm.nih.gov/gds (accessed on 6 March 2021) was also analyzed by using “limma” package: http://www.bioconductor.org/packages/release/bioc/html/limma.html (accessed on 6 March 2021) of the R language with |logFC| > 0.5 and *p* < 0.05 as thresholds for differential analysis of bladder cancer microarray data GSE65635 and GSE40355. There were 12 samples in microarray data GSE65635, including 4 normal samples and 8 bladder cancer samples. There were 24 samples in microarray data GSE40355, including 8 normal samples and 16 bladder cancer samples. Human lncRNA names were obtained from GENCODE, followed by finding the intersection of significantly differential genes and lncRNA names. A Venn diagram was drawn to screen out the lncRNAs from intersection. LncRNA expression trends were collected in Ualcan, and the key lncRNA was determined by comparing the expression trends and combining with the existing literature. The possible downstream miR of the key lncRNA was discovered by starBase and their binding sites were obtained. The databases TargetScan (cumulative weighted context++ score < 0), DIANA TOOLS (miTG score > 0.6), microRNA (conservation > 0.65, energy < −14, Mirsvr_score < −0.65), and mirDIP (integrated score > 0.1) was applied to predict downstream genes of miR. Intersection of downstream genes with significantly differential genes was taken to find critical downstream genes. The relevant genes of the critical downstream gene were predicted in GeneMANIA: http://genemania.org/ (3 April 2021), followed by construction of a protein-protein interaction (PPI) network. The most core genes in PPI network (Table 1) were chosen as the key gene, and the binding sites of the miR to the gene were predicted by TargetScan.

### 2.4. Fluorescence In Situ Hybridization (FISH)

SNHG1 cDNA fragments were amplified from the SNHG1 plasmid as templates by utilizing high fidelity DNA polymerase (Takara, Kyoto, Japan). Based on this template, fluorescein-labeled lncTCF7 FISH probe DNA was prepared with fluorescein-12-dUTP (Roche, Mannheim, Germany) and Klenow DNA polymerase (Vazyme, Nanjing, China) as per the manufacturer’s protocol. Then, 4-µm frozen sections were made from bladder cancer tissues and adjacent normal tissues. Subsequent to 5-min immersion in proteinase K (MCE, NJ, USA), the slides were washed twice in 2× saline sodium citrate (SSC) (Merck KGaA, Darmstadt, Germany). The FISH hybridization solution encompassing 30 ng/µL lncTCF7 FISH probe DNA (Beijing Dingguo Changsheng Biotechnology Co., Ltd., Beijing, China) was dripped onto the tissue sections before 16-h incubation at 37 °C. The slides were then washed in 0.4 × SSC/0.001% NP-40 (Merck KGaA, Darmstadt, Germany) for 5 min at 56 °C, followed by another 2-min washing in 0.4 × SSC/0.001% NP-40. After being dripped with 4′,6-Diamidino-2-Phenylindole (DAPI)-encompassing sealing agent, the slide was mounted and observed under a fluorescence microscope (Olympus, Tokyo, Japan). 

### 2.5. Cell Incubation

Human normal urothelial cell line SV-HUC-1 (ATCC^®^ CRL-9520), and bladder cancer cell lines [5637 (ATCC^®^ HTB-9), T24 (ATCC^®^ HTB-4™), SW780 (ATCC^®^ CRL-2169™), and UM-UC-3 (ATCC^®^ CRL-1749™) were obtained from American Type Culture Collection (ATCC, Manassas, VA, USA). The medium used for SV-HUC-1 was ATCC-formulated F-12K (Gibco, Thermo Fisher Scientific, Waltham, MD, USA) Medium (Catalog No. 30-2004) encompassing 10% fetal bovine serum (FBS, ATCC 30-2020). The medium used for 5637 was ATCC-formulated RPMI-1640 Medium (Gibco, Thermo Fisher Scientific, Waltham, MD, USA) (ATCC 30-2001) encompassing 10% FBS. The medium for T24 was ATCC-formulated McCoy’s 5A Medium Modified (Gibco, Thermo Fisher Scientific, Waltham, MD, USA) (Catalog No. 30-2007) with 10% FBS. The medium for SW780 was ATCC-formulated Leibovitz’s L-15 Medium (Gibco, Thermo Fisher Scientific, Waltham, MD, USA) (Catalog No. 30-2008) with 10% FBS. The medium for UM-UC-3 was ATCC-formulated Eagle’s Minimum Essential Medium (Gibco, Thermo Fisher Scientific, Waltham, MD, USA) (Catalog No. 30-2003) with 10% FBS. All media for cell lines encompassed 100 μg/mL streptomycin (MCE, N.J, USA) and 100 U/mL penicillin (MCE, NJ, USA). Cell culture was performed at 37 °C with 5% CO_2_. The media were positioned in humid air and replaced every 2–3 days according to the growth of cells. Cells were subcultured when 80–90% of the culture plate was covered by cells. Cells were utilized when they reached the logarithmic growth stage.

### 2.6. Cell Transfection

Lentiviruses expressing specific targeted knockdown SNHG1 [short hairpin-SNHG1 (sh-SNHG1)] and MDM2 (sh-MDM2) sequences and a scramble shRNA (sh-NC; control shRNA) were constructed by GenePharma (Shanghai, China) (Table 2). Lentiviruses overexpressing SNHG1 (oe-SNHG1), MDM2 (oe-MDM2), and PPARγ (oe-PPARγ), oe-NC, Inhibitor NC, and miR-9-3p inhibitor were obtained from GenePharma. Transfection was implemented using Lipofectamine 3000 (Invitrogen, Carlsbad, CA, USA).

### 2.7. Reverse Transcription Quantitative Polymerase Chain Reaction (qRT-PCR)

Subsequent to isolation using RNeasy Mini Kit (Qiagen, Valencia, CA, USA), total RNA underwent reverse transcription to generate cDNA using First Strand cDNA Synthesis Kit (RR047A, Takara). For the detection of miR, the cDNA was obtained by reverse transcription using the miRNA First Strand cDNA Synthesis (Tailing Reaction) kit (B532451-0020, Sangon, Shanghai, China). qRT-PCR reactions were performed using SYBR^®^ Premix Ex TaqTM II (Perfect Real Time) kit (DRR081, Takara) on real-time fluorescence quantitative PCR instrument (ABI 7500, Applied Biosystems, Foster City, CA, USA). The universal reverse primers for miR and the upstream primers for U6 internal reference were provided in the miRNA First Strand cDNA Synthesis (Tailing Reaction) kit, and the other primers were synthesized by Sangon (Table 3). After recording the Ct value of each well, the relative expression of mRNAs or miR was calculated using the 2^−ΔΔCt^ method by normalizing to U6 expression.

### 2.8. Cell Counting Kit (CCK)-8 Assay

The transfected T24 and 5367 cells (the expression of SNHG1 was clearly highest in T24 and compared with sw780 cells, 5637 cells had better culture characteristics in our experiment) were resuspended and seeded in 96-well plates at 2 × 10^3^/100 µL/well. Cell viability was evaluated by CCK-8 (Dojindo Laboratories, Kumamoto, Japan) method at 0, 24, 48, 72 and 96 h after seeding. The 10 µL CCK-8 solution was supplemented in each test for 4-h incubation before absorbance measurement at 450 nm with a microplate reader.

### 2.9. 5-Ethynyl-2′-Deoxyuridine (EdU) Assay

The cells to be tested were seeded in 24-well plates with three duplicated wells set for cells in each group. EdU (Invitrogen) was supplemented to the medium to achieve a concentration of 10 µmol/L. The medium was discarded subsequent to 2-h culture. Cells received 15-min phosphate buffer saline (PBS) encompassing 4% paraformaldehyde fixing at ambient temperature before 20-min incubation at ambient temperature with PBS encompassing 0.5% Triton-100. Each well was supplemented with 100 µL dye solution before 30-min culture in the dark at ambient temperature. DAPI was added for 5-min nuclear staining. After sealing, 6–10 fields of view were randomly observed under a fluorescence microscope (FM-600, Shanghai Pudan Optical Instrument Co., Ltd., Shanghai, China), and the number of positive cells in each field was recorded.

### 2.10. Flow Cytometry

Subsequent to 48-h transfection, the cell concentration was changed to 1 × 10^6^ cells/mL. After cell fixing with 70% precooled ethanol solution at 4 °C, 100 μL cell suspension (no less than 1 × 10^6^ cells/mL) was resuspended in 200 μL binding buffer. Subsequently, 15-min cells staining was implemented with 10 μL Annexin V-fluoresceinisothiocyanat and 5 μL propidium iodide at ambient temperature under dark conditions. After 300 μL of binding buffer was added, apoptosis (T24 and 5367 cell lines) was assessed on a flow cytometer at excitation wavelength of 488 nm (2 × 10^4^ cells each time).

### 2.11. Western Blot Analysis

Subsequent to trypsin treatment, cells were lysed with enhanced radio-immunoprecipitation assay (RIPA) lysis encompassing protease inhibitors (BOSTER, Wuhan, Hubei, China), followed by estimation of protein concentration using Bicinchoninic Acid (BCA) Protein Quantification Kit (BOSTER). Proteins underwent separation by 10% sodium dodecyl sulfate polyacrylamide gel electropheresis (SDS-PAGE). Then, the separated proteins were electroblotted into a polyvinylidene fluoride (PVDF) membrane which was sealed by 5% bovine serum albumin to block nonspecific binding. Overnight cell incubation was conducted at 4 °C after supplementation with primary rabbit antibodies (Abcam, Cambridge, UK) to Cleaved caspase-3 (ab49822, 1:500), Bcl-2-Associated X (Bax, ab32503, 1:1000), B-cell lymphoma-2 (Bcl-2, ab196495, 1:500), MDM2 (ab226939, 1:3000), PPARγ (ab45036, 1:500), Ubiquitin (ab7780, 1:2000), and β-actin (ab8227, 1:500). Then, horseradish peroxidase-tagged goat anti-rabbit secondary antibodies (ab205719, 1:2000, Abcam) were supplemented for 1-h membrane incubation at 4 °C. After development in ECL working fluid (EMD Millipore Corporation, Billerica, MA, USA), the bands in Western blot images were quantified by Image J analysis software by normalizing to β-actin.

### 2.12. RNA Pull Down

Cells were transfected with biotinylated wild type (WT) miR-9-3p and mutant type (MUT) miR-9-3p (50 nM each). After 48 h of transfection, 10-min cell incubation was implemented with specific cell lysis (Ambion, Austin, Texas, TX, USA). Then, 3-h lysate incubation was conducted with M-280 streptavidin magnetic beads (Sigma, St. Louis, MO, USA) pre-coated with RNase-free and yeast tRNA at 4 °C before two cell washes in cold lysis and qRT-PCR detection of SNHG1 expression.

### 2.13. RNA Immunoprecipitation (RIP) Assay

The binding of miR-9-3p to MDM2 was detected by RIP kit (Millipore, Temecula, CA, USA). Briefly, 5-min cell lysing was implemented in an ice bath with equal volume of RIPA lysis (P0013B, Beyotime, Shanghai, China), and supernatant was removed subsequent to 10-min centrifugation at 14,000× *g* rpm and 4 °C. A portion of the cell extract was applied as input, and a portion was co-precipitated with antibody. RNA extraction was implemented by treating samples with proteinase K for subsequent qRT-PCR detection of MDM2. Antibodies used for RIP were as follows: rabbit anti-Argonaute 2 (AGO2) (1:100, ab32381, Abcam) was mixed at ambient temperature for 30 min, and rabbit anti-human Immunoglobulin G (IgG; 1:100, ab109489, Abcam) was applied as a normal control (NC).

### 2.14. Dual Luciferase Reporter Gene Assay

The synthesized MDM2 3′ untranslated region (UTR) gene fragment MDM2-WT and the MDM2-MUT mutated at the binding site were constructed into a pMIR-reporter plasmid (Beijing Huayueyang Biotechnology, Beijing, China). Luciferase reporter plasmids were co-transfected with miR-9-3p into HEK293T cells (Shanghai Beinuo Biotechnology, Shanghai, China). Then 48 h subsequent to transfection, cells were lysed, and detected using a luciferase detection kit (K801-200; Biovision, Mountain View, CA, USA).

### 2.15. Immunoprecipitation (IP)

T24 cells were lysed in lysis buffer [mixture of 50 mM Tris-HCl (pH 7.4), 150 mM NaCl, 10% glycerol, 1 mM ethylene diamine tetraacetic acid, 0.5% NP-40, and protease inhibitor], and cell debris was cleared by centrifugation. After the concentration of lysis was measured by BCA, the same amount of protein was taken from oe-NC/OE-MDM2 group and replenished to the same volume with cell lysate. Afterwards, 1 μg anti-MDM2 (ab226939, 1:100, Abcam), PPARγ (ab45036, 1:100, Abcam) and 15 μL protein A/G beads (Santa Cruz Biotechnology, Santa Cruz, CA, USA) were added for 2-h incubation. Subsequent to three washes with cell lysis, beads were collected by centrifugation, added into an equal volume of reductive loading buffer, and boiled at 100 °C for 5 min. Subsequent to SDS-PAGE, samples were electroblotted to PVDF membranes (Millipore), and then analyzed by immunoblotting.

### 2.16. Subcutaneous Tumorigenesis Model in Nude Mice

Healthy nude mice aged 6–8 weeks (Beijing Institute of Pharmacology, Chinese Academy of Medical Sciences, Beijing, China) were bred in a specific pathogen-free animal laboratory with 60–65% humidity at 22–25 °C. They were fed in separate cages under 12:12-h light-dark cycle with food and water available ad libitum. The experiment was started one week after acclimation, and the health status of nude mice was observed before the experiment. Approximately 2 ×  10^6^ 5637 cells were suspended in 200 μL PBS, and then subcutaneously injected into the left or right hindlimbs of nude mice (antagomir used). At 28 days subsequent to injection, mice were euthanized, followed by measurement and weighing of tumors.

### 2.17. Statistical Analysis

All measurement data were manifested as mean ± standard deviation and analyzed by SPSS 21.0 software (IBM, Armonk, NY, USA), with *p* < 0.05 as a level of statistical significance. If data conformed to normal distribution and homogeneity of variance, data within groups were compared by paired *t* test, while data between two groups were compared by unpaired *t* test. Comparisons among multiple groups were performed using one-way analysis of variance (ANOVA) or repeated measures ANOVA. Intra-group pairwise comparison was performed using a post-hoc test. Rank sum test was performed if data did not conform to normal distribution or homogeneity of variance. Kaplan–Meier was adopted to calculate patient survival curves, and log-rank was utilized to analyze patient survival differences.

## 3. Results

### 3.1. SNHG1 Was over Expressed and Associated with Poor Prognosis in Bladder Cancer

The BLCA data of TCGA database were analyzed by GEPIA to obtain 6597 significantly differential genes (|logFC| > 0.5, *p* < 0.05) (Figure 1A). Then, R language was employed for difference analysis of microarray data GSE65635 and GSE40355 in GEO database to obtain 4283 and 8065 significantly differential genes, respectively (|logFC| > 0.5, *p* < 0.05). Then, 17,937 human lncRNA names were obtained from GENCODE, which were intersected with differential genes. It was found that only DIO3OS and SNHG1 were significantly differential lncRNAs in bladder cancer (Figure 1B). Analysis of TCGA database data by Ualcan revealed that DIO3OS was significantly underexpressed in bladder cancer (*p* = 1.949 × 10^−5^; Figure 1C), while SNHG1 was significantly overexpressed in bladder cancer (*p* = 3.889 × 10^−11^; Figure 1C). Moreover, the difference of SNHG1 was significantly higher than that of DIO3OS. There was literature indicating the upregulation of SNHG1 in thyroid cancer [14], non-small cell lung cancer (NSCLC) [15], colorectal cancer [16,17], but SNHG1 was not studied in bladder cancer. Further detection by qRT-PCR assay also found that SNHG1 was highly expressed in bladder cancer tissues (Figure 1D). RNA-FISH showed high SNHG1 expression in bladder cancer tissues compared with adjacent normal tissues as well (Figure 1E). It was also observed that patients with high expression of SNHG1 had worse prognosis (Figure 1F). Therefore, high SNHG1 expression was associated with poor prognosis of patients with bladder cancer.

### 3.2. SNHG1 Promoted Bladder Cancer Cell Proliferation by Inhibiting Apoptosis

To further examine the regulatory role of SNHG1 in bladder cancer, we selected one normal urothelial cell line SV-HUC-1 and four bladder cancer cell lines (5637, T24, SW780, and UM-UC-3). As shown in Figure 2A, SNHG1 expression was increased in cancer cells compared with SV-HUC-1 cells. Subsequent experiments were conducted on T24 cells with higher SNHG1 expression and 5637 cells with lower SNHG1 expression. After silencing SNHG1 in T24 cells (Figure 2B), the shRNA with the highest silencing efficiency was selected for subsequent experiments. CCK-8 and EdU assays showed clearly that the viability and proliferation of T24 cells were inhibited by treatment with sh-SNHG1 (Figure 2C,D). The apoptotic rate of T24 cells elevated significantly (Figure 2E), accompanied by prominent increase of Cleaved caspase-3 and Bax expression and remarkable decline of Bcl-2 expression (Figure 2F), after silencing SNHG1. This suggested that silencing SNHG1 could trigger the inhibition of cell proliferation and promotion of cell apoptosis in bladder cancer. Further experiments in 5637 cells manifested that overexpression of SNHG1 in these cells (Figure 3A), noteworthy enhanced viability (Figure 3B), and proliferation (Figure 3C), diminished apoptosis (Figure 3D), and reduced the expression of Cleaved caspase-3 and Bax but elevated Bcl-2 expression (Figure 3E). Collectively, SNHG1 upregulation promoted the proliferation of bladder cancer cells.

### 3.3. SNHG1 Promoted Bladder Cancer Cell Tumorigenesis In Vivo

Furthermore, a subcutaneous tumorigenic model was established in nude mice to detect the tumorigenic ability of bladder cancer cells in vivo. qRT-PCR depicted that SNHG1 expression was appreciably decreased in mice treated with sh-SNHG1 (Figure 4A). In addition, the growth rate and weight of tumors were appreciably decreased after silencing SNHG1 (Figure 4B). FISH experiment illustrated that SNHG1-silenced mice had distinct decline of SNHG1 expression (Figure 4C). On the contrary, overexpression of SNHG1 (Figure 4D) contributed to the appreciable elevation of the growth rate and weight of tumors (Figure 4E) and SNHG1 expression in tumors (Figure 4F). In summary, SNHG1 overexpression promoted bladder cancer cell tumorigenesis in vivo.

### 3.4. SNHG1 Silencing Suppressed Bladder Cancer Cell Proliferation and Tumorigenesis by Binding to MiR-9-3p

Then, we explored the downstream miR of SNHG1 in bladder cancer. RNA-FISH (Figure 1E) showed that SNHG1 was localized in the cytoplasm, suggesting that SNHG1 may be involved in the process of bladder cancer by affecting miR. The starBase website predicted that SNHG1 could bind to miR-9-3p (Figure 5A). A previous study has reported that miR-9-3p expression is down-regulated in bladder cancer [10], but the potential regulatory mechanisms need further detecting. qRT-PCR revealed that miR-9-3p expression in bladder cancer tissues was significantly lower than that in their matched nontumor adjacent tissues (Figure 5B). The expression of miR-9-3p was negatively correlated to the expression of SNHG1 in bladder cancer tissues (Figure 5C). Meanwhile, it was verified by RNA pull-down that SNHG1 indeed bound to miR-9-3p (Figure 5D). In addition, silencing SNHG1 in T24 cells prominently increased miR-9-3p expression (Figure 5E), while overexpressing SNHG1 in 5637 cells severely declined miR-9-3p expression (Figure 5F).

The effect of SNHG1 binding to miR-9-3p on bladder cancer cells was further examined. Silencing SNHG1 alone resulted in decrease of SNHG1 expression. Besides, silencing SNHG1 alone reduced cell proliferation, increased apoptotic rate, elevated Cleaved caspase-3 and Bax expression, and caused a decline in Bcl-2 expression in T24 cells; this contrasted with treatment with miR-9-3p inhibitor alone. However, co-treatment with sh-SNHG1 and miR-9-3p inhibitor reversed the effect of sh-SNHG1 or miR-9-3p inhibitor alone, when the miR-9-3p inhibitor was used in T24 cells. However, co-treatment with sh-SNHG1 and miR-9-3p inhibitor reversed the effect of sh-SNHG1 or miR-9-3p inhibitor alone in cck-8 assay (Figure 6A–E). Simultaneously, in vivo experiments showed that the tumorigenic ability of bladder cancer cells in vivo was diminished by treatment with sh-SNHG1 alone and elevated by treatment with miR-9-3p inhibitor alone, which was neutralized by co-treatment with sh-SNHG1 and miR-9-3p antagomir (Figure 6F,G). Conclusively, silencing SNHG1 bound to miR-9-3p can inhibit bladder cancer cell proliferation and tumorigenesis.

### 3.5. Silencing SNHG1 Decreased MDM2 Expression through MiR-9-3p

Subsequently, the downstream target genes of miR-9-3p were investigated. The 3615, 2399, 270 and 2387 downstream genes of miR-9-3p were respectively predicted in TargetScan, DIANATOOLS, microRNA and mirDIP, and then were intersected. The intersecting results were compared with the differential genes in bladder cancer obtained by GEPIA, which screened out 21 significantly differential downstream genes of miR-9-3p (Figure 7A). By constructing the PPI network through GeneMANIA, we found that MDM2 had the highest core degree in the PPI network and was double the core degree of second-ranked genes (Figure 7B, Table 1). The binding site of miR-9-3p in MDM2 3′UTR was predicted using TargetScan (Figure 7C). Nevertheless, previous studies detected that MDM2 was highly expressed as a proto-oncogene in bladder cancer [11,18]. miR-9-3p may be involved in the progression of bladder cancer by inhibiting the expression of MDM2, which was further verified by experiments. Firstly, AGO2 pulled down MDM2 in RIP experiments (Figure 7D). Dual luciferase reporter assay manifested that miR-9-3p mimic markedly inhibited the luciferase activity of MDM2 WT but had no obvious effect on the luciferase activity of MDM2 MUT, suggesting that miR-9-3p bound to the 3′UTR of MDM2 (Figure 7E).

After sh-SNHG1 and miR-9-3p inhibitor were co-transfected into T24 cells, MDM2 expression was evaluated by qRT-PCR and Western blot analysis. As displayed in Figure 7F, G, MDM2 expression was noticeably decreased after silencing SNHG1 alone, and observably increased after transfection with miR-9-3p inhibitor alone, which was normalized by co-treatment with sh-SNHG1 and miR-9-3p inhibitor. It was suggested that silencing of SNHG1 inhibited MDM2 expression through binding to miR-9-3p.

### 3.6. SNHG1 Silencing Inhibited Bladder Cancer Progression by Downregulating MDM2

We observed that the expression of MDM2 was overexpressed in SNHG1-silenced T24 cells. From qRT-PCR results, the expression of SNHG1 and MDM2 was substantially diminished and miR-9-3p expression was significantly enhanced after silencing SNHG1 alone. MDM2 expression was appreciably increased after overexpressing MDM2 alone, and silencing SNHG1 negated the effect of overexpressing MDM2 (Figure 7H). As described in Figure 7I,J, CCK-8 and EdU assays exhibited that silencing SNHG1 appreciably reduced but overexpressing MDM2 increased cell proliferation, and overexpressing MDM2 could reverse the effect of silencing SNHG1 on cell proliferation (Figure 7I–J). To sum up, SNHG1 silencing suppressed the proliferation of bladder cancer cells by decreasing MDM2 expression through miR-9-3p.

Similar results were confirmed in vivo. The expression of SNHG1, miR-9-3p, and MDM2 was detected by qRT-PCR. As presented in Figure, SNHG1 and MDM2 expression was strikingly declined and miR-9-3p expression was remarkably elevated after silencing SNHG1. MDM2 expression was prominently promoted after overexpressing MDM2, and overexpressing MDM2 could reverse the effect of silencing SNHG1 on MDM2 expression (Figure 7K). Moreover, after silencing SNHG1 alone, tumor growth was repressed, and after overexpressing MDM2 alone, tumor growth was promoted. Overexpression of MDM2 abrogated the effect of silencing SNHG1 on tumor growth (Figure 7L). In summary, silencing SNHG1 decreased the tumorigenesis of bladder cancer cells in vivo by decreasing MDM2 expression through miR-9-3p.

### 3.7. Silencing SNHG1 Upregulated PPARγ through MDM2

It has been documented that after addition of EGFR, MDM2 can bind to PPARγ and regulate the ubiquitination of PPARγ protein in colon cancer, and that MDM2 silencing can increase the level of PPARγ [12], which is further verified in bladder cancer. Firstly, through IP experiments in T24 cells, it was found that MDM2 combined with PPARγ. Meanwhile, the combination of PPARγ to MDM2 can also be proved (Figure 8A). Further, after screening out the MDM2 silencing sequence (Figure 8B, sh-MDM2-3 was selected for subsequent experiments), we found that PPARγ ubiquitination decreased and PPARγ expression increased after silencing MDM2 with the addition of EGFR (Figure 8C), which was opposite after overexpressing MDM2 (Figure 8D). MDM2 was overexpressed after silencing SNHG1 in T24 cells with the addition of EGFR. Western blot analysis showed that MDM2 expression was prominently decreased and PPARγ expression was strongly elevated after silencing SNHG1 alone, which was an opposite effect to overexpressing MDM2 alone. Silencing SNHG1 annulled the effect of overexpressing MDM2 on MDM2 and PPARγ expression (Figure 8E). The above results suggested that MDM2 reduced PPARγ expression by inducing PPARγ ubiquitination, while silencing of SNHG1 elevated PPARγ expression through downregulating MDM2.

## 4. Discussion

As one of the most prevalent genitourinary cancers with high mortality on a global scale, bladder cancer currently can be treated by local or systemic immunotherapy, radiotherapy, chemotherapy, and endoscopic and open surgery [19]. However, the curative effect of such therapies is limited because of recurrence or distant spread [20]. Moreover, lncRNAs has emerged as a modulator in the complexity of bladder cancer [21]. Consequently, this research investigated the mechanism of SNHG1 in bladder cancer with the involvement of miR-9-3p. Notably, the present study provided evidence that SNHG1 promotes MDM2 expression by binding to miR-9-3p to promote PPARγ ubiquitination and downregulate PPARγ expression, thereby resulting in elevation of bladder cancer cell proliferation in vitro and tumorigenesis in vivo.

Initially, data from our project and online database unraveled that SNHG1 was highly expressed in bladder cancer tissues and cells. Additionally, when SNHG1 was silenced in bladder cancer cells and mice, cell proliferative capacity was depressed but cell apoptosis was accelerated in vitro, and tumorigenesis was inhibited in vivo. Importantly, SNHG1 has emerged as a novel oncogenic lncRNA in various cancers, including esophageal, colorectal, prostate, gastric, liver, and lung cancers, inducing cell proliferative, metastatic, migratory and invasive capacities of cancer cells [6]. Consistently, Lu et al. observed that SNHG1 expression was strikingly high in NSCLC tissues and cells, and that SNHG1 silencing decreased tumor volumes in mice and reduced NSCLC cell proliferation, invasion and migration [22]. Meanwhile, data collected by Bai et al. found that SNHG1 expression was upregulated in colorectal cancer cells, and that ectopically expressed SNHG1 could enhance cell migratory, proliferative, and invasive capacities in vitro and led to tumor growth [23,24]. Another study uncovered that SNHG1 silencing contributed to decline of tumor growth of breast cancer in vivo [25]. These findings indirectly supported the tumor-promoting potential of SNHG1 in bladder cancer by enhancing cell proliferation and tumor growth and reducing apoptosis.

It is well-recognized that lncRNAs may function as endogenous sponges to regulate miRNA function in diseases [26]. For example, prior research showed that SNHG1 could bind to miR-204 to inhibit it, thus promoting migratory, and invasive abilities but repressing apoptosis in esophageal squamous cell cancer [27]. These findings indirectly confirmed the binding relationship between SNHG1 and miR-9-3p in bladder cancer cells which was observed by our study. Further investigations of our study identified that miR-9-3p inhibition led to increase of cell proliferation and decrease of apoptosis in vitro and promotion of tumorigenesis in vivo and reversed the effect of SNHG1 silencing in bladder cancer. Similarly, the research conducted by Cai et al. clarified that in bladder cancer, miR-9-3p overexpression triggered repression of cell viability, migration, and invasion, induction of cell apoptosis in vitro, and inhibition in vivo tumor growth and metastasis [10]. Notably, another study illustrated that miR-9-3p exerted tumor-suppressive effect on hepatocellular carcinoma by depressing hepatocellular carcinoma cell proliferation [28], which was in line with our results. Hence, these results confirmed that SNHG1 overexpression promoted bladder cancer progression by binding to miR-9-3p.

It is well-established that miRs inhibit expression of target genes at post-transcriptional level by targeting the 3′UTR of mRNA [29]. As previously reported, miR-9-3p targeted HBGF-5 to function as a tumor suppressor in hepatocellular carcinoma [30]. Moreover, in our study, TargetScan website predicted the binding sites between miR-9-3p and MDM2 3′UTR, and then the targeting relationship between miR-9-3p and MDM2 was verified by dual luciferase reporter gene assay. In the subsequent experiments, we found that MDM2 bound to PPARγ and downregulated PPARγ by inducing PPARγ ubiquitination, which was similar to the results observed by Xu et al. [12]. Our experiments show that Bcl-2 and Bax were also related to a certain extent, which is worth further in-depth study. Furthermore, our data elaborated that MDM2 ectopic expression neutralized the inhibitory effect of SNHG1 silencing on cell proliferation in vitro and tumor growth in vivo and the promoting effect of SNHG1 silencing on cell apoptosis in vitro in bladder cancer, suggesting the oncogenic role of MDM2 in bladder cancer. Similarly, a prior study uncovered that inhibition of MDM2 exerted tumor-suppressive effects on bladder cancer by decreasing cell invasive, proliferative, and migratory capacities [11]. Accordingly, PPARγ activation gave rise to inhibition of proliferation of 9 bladder cancer cell lines [31] All in all, the SNHG1/miR-9-3p/MDM2/PPARγ axis was involved in bladder cancer progression.

Collectively, this study provides evidence that SNHG1 upregulation promoted cell proliferation but depressed cell apoptosis in bladder cancer via MDM2-inhibited PPARγ by binding to miR-9-3p (Figure 9). Thus, this finding offers a fresh molecular insight that might be utilized in new therapy development for bladder cancer. However, further studies are required on the mechanism of PPARγ in bladder cancer.

## Figures and Tables

**Figure 1 cancers-14-04740-f001:**
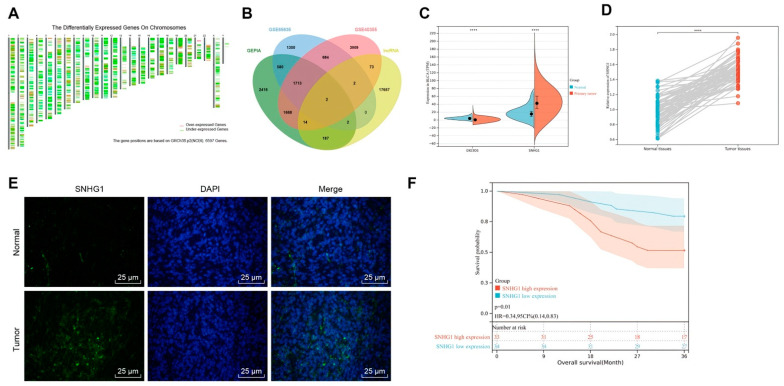
SNHG1 is upregulated in bladder cancer tissues in association with poor prognosis of patients with bladder cancer. (**A**) GEPIA: http://gepia2.cancer-pku.cn/ (accessed on 6 June 2021) analysis of the location map of differentially expressed genes obtained from BLCA data in TCGA database: https://portal.gdc.cancer.gov/ (accessed on 6 June 2021). Chromosomes 1-22, X and Y are shown in turn from left to right. (**B**) Venn diagram of the intersection of significantly differential genes obtained from GEPIA analysis, GSE65635, and GSE40355 and human lncRNA obtained from GENCODE: https://www.gencodegenes.org/human/ (accessed on 7 June 2021) (intersecting genes were DIO3OS and SNHG1). (**C**) UALCAN: http://ualcan.path.uab.edu/index.html (accessed on 7 June 2021) analysis of TCGA database showing that DIO3OS expression was significantly low in bladder cancer, while SNHG1 expression was significantly high in bladder cancer. (**D**) qRT-PCR detection of SNHG1 expression in 40 cases of bladder cancer and adjacent tissues. (**E**) RNA-FISH detection of SNHG1 expression in 40 cases of bladder cancer and adjacent tissues (400×, scale bar = 25 μm). (**F**) Kaplan–Meier analysis of SNHG1 and patient survival curve. **** *p* < 0.0001.

**Figure 2 cancers-14-04740-f002:**
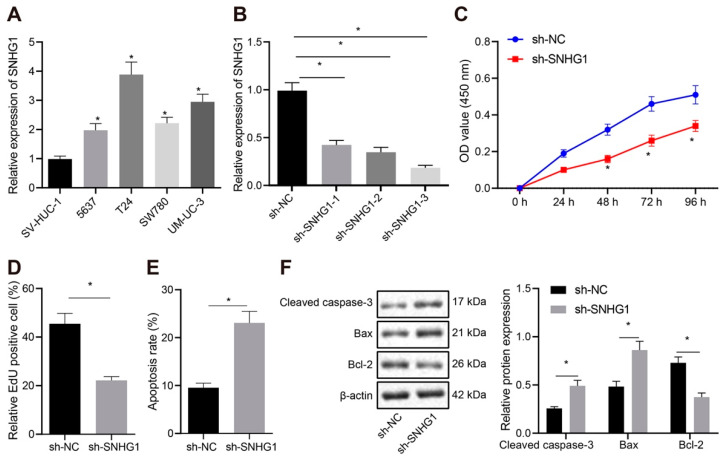
Silencing SNHG1 inhibits bladder cancer cell proliferation and promotes cell apoptosis. (**A**) qRT-PCR detecting the expression of SNHG1 in one normal urothelial cell line SV-HUC-1 and four bladder cancer cell lines (5637, T24, SW780, and UM-UC-3), and selection of T24 cells with the highest expression and 5367 cells with the lowest expression for subsequent experiments. T24 cells were transfected with si-NC or si-SNHG1. (**B**) qRT-PCR determination of the silencing efficiency of three sh-SNHG1 in T24 cells, and screening out of the highest silencing efficiency sh-SNHG1-3. (**C**) CCK-8 assay of the change of T24 cell viability. (**D**) The changes of T24 cell proliferation detected by EdU assay. (**E**) Flow cytometry analysis of the changes of T24 cell apoptosis. (**F**) Western blot analysis of the expression of Cleaved caspase-3, Bax, and Bcl-2 in T24 cells. * *p* < 0.05. The experiment was repeated three times. Data were presented as mean ± standard deviation and compared by one-way analysis of variance (ANOVA), followed by Tukey’s post hoc test.

**Figure 3 cancers-14-04740-f003:**
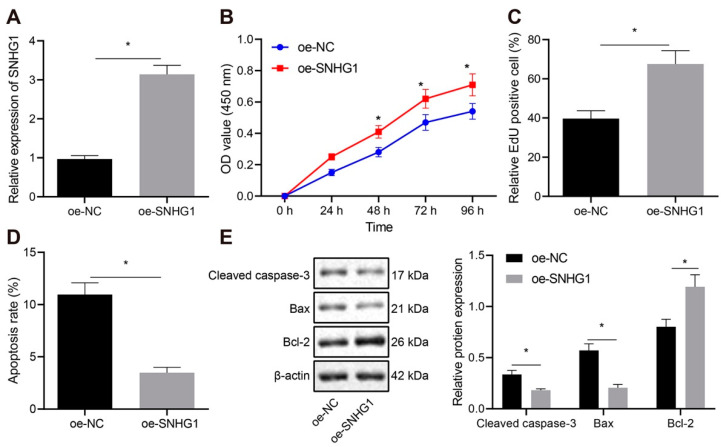
Overexpression of SNHG1 stimulates bladder cancer cell proliferation and represses cell apoptosis. (**A**) qRT-PCR determination of the overexpression efficiency of oe-SNHG1 in 5637 cells. (**B**) CCK-8 assay of the change of 5637 cell viability. (**C**) The changes of 5637 cell proliferation detected by EdU assay. (**D**) Flow cytometry analysis of the changes of 5637 cell apoptosis. (**E**) Western blot analysis of the expression of Cleaved caspase-3, Bax, and Bcl-2 in 5637 cells. * *p* < 0.05. The experiment was repeated three times. Data were presented as mean ± standard deviation and compared by *t* test. OE: over expression.

**Figure 4 cancers-14-04740-f004:**
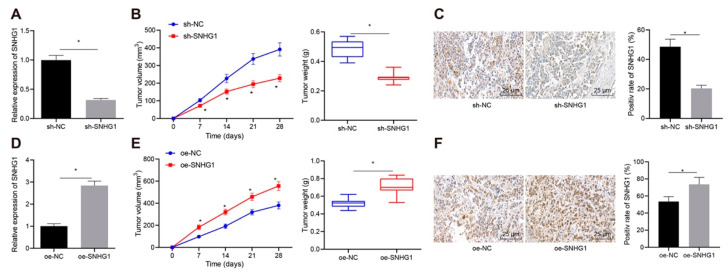
SNHG1 silencing represses bladder cancer cell tumorigenesis in vivo. (**A**) qRT-PCR to detect the expression of SNHG1 in tumors. (**B**) Tumor weight. (**C**) FISH to detect the expression of SNHG1 in tumors (400×, scale bar = 25 μm). The 5637 cells stably transfected with oe-SNHG1 were subcutaneously injected into the axilla of nude mice. (**D**) qRT-PCR to detect the expression of SNHG1 in tumors. (**E**) Tumor weight. (**F**) FISH to detect the expression of SNHG1 in tumors (400×, scale bar = 25 μm). * *p* < 0.05. N = 10 mice in each group. Data were presented as mean ± standard deviation and compared by *t* test or repeated measures ANOVA with Tukey’s post hoc test.

**Figure 5 cancers-14-04740-f005:**
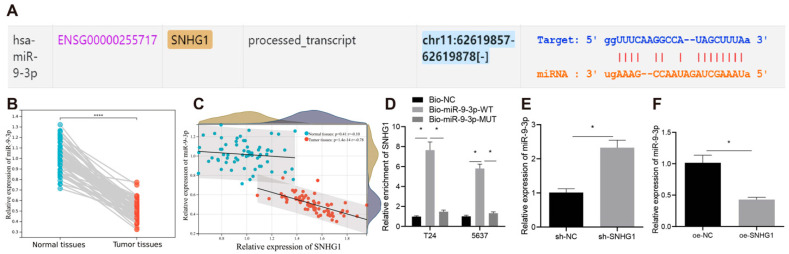
SNHG1 binds to and downregulates miR-9-3p. (**A**) starBase: http://starbase.sysu.edu.cn/ (5 July 2021) website predicting that SNHG1 bound to miR-9-3p. (**B**) qRT-PCR detecting the expression of miR-9-3p in 40 bladder cancer and adjacent normal tissues. (**C**) Correlation analysis of the expression of SNHG1 and miR-9-3p. (**D**) RNA pull-down assay detecting the binding relationship between SNHG1 and miR-9-3p. (**E**) qRT-PCR detection of the expression of miR-9-3p after silencing SNHG1 in T24 cells. (**F**) qRT-PCR detection of the expression of miR-9-3p after overexpressing SNHG1 in 5637 cells. * *p* < 0.05. The experiment was repeated three times. Data were presented as mean ± standard deviation and compared by *t* test or one-way analysis of variance (ANOVA), followed by Tukey’s post hoc test.

**Figure 6 cancers-14-04740-f006:**
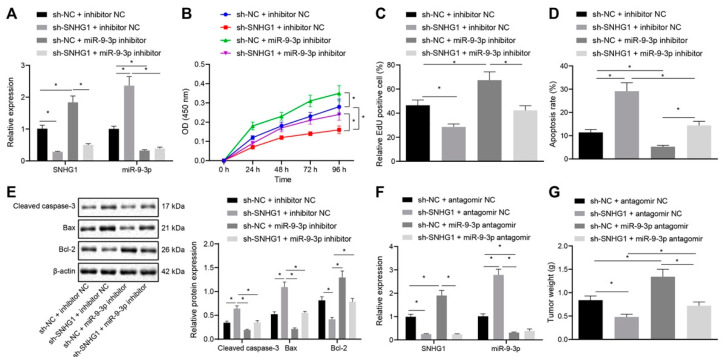
SNHG1 silencing binds to miR-9-3p to repress bladder cancer cell proliferation and tumorigenesis. (**A**) SNHG1 and miR-9-3p expression in T24 cells measured by qRT-PCR. (**B**) CCK-8 assay of the change of T24 cell viability. (**C**) The changes of T24 cell proliferation detected by EdU assay. (**D**) Flow cytometry analysis of the changes of T24 cell apoptosis. (**E**) Western blot analysis of the expression of Cleaved caspase-3, Bax, and Bcl-2 in T24 cells. The stably transfected T24 cells were subcutaneously injected into the axilla of nude mice. (**F**) SNHG1 and miR-9-3p expression in mice measured by qRT-PCR. (**G**) Tumor weight. * *p* < 0.05. N = 10 mice in each group. The cell experiment was repeated three times. Data were presented as mean ± standard deviation and compared by one-way analysis of variance or repeated measures ANOVA with Tukey’s post hoc test.

**Figure 7 cancers-14-04740-f007:**
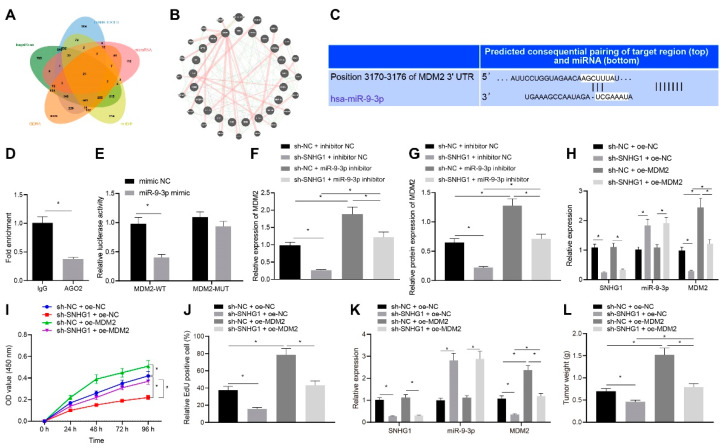
SNHG1 silencing decreases MDM2 expression via miR-9-3p to repress bladder cancer cell proliferation and tumorigenesis. (**A**) The downstream genes of miR-9-3p predicted by TargetScan: http://www.targetscan.org/vert_71/ (accessed on 8 August 2021), DIANA TOOLS: http://diana.imis.athena-innovation.gr/DianaTools/ (accessed on 8 August 2021), microRNA: http://www.microrna.org/microrna/home.do (accessed on 8 August 2021), and mirDIP: http://ophid.utoronto.ca/mirDIP/ (accessed on 8 August 2021). (**B**) The differential genes in bladder cancer obtained by GEPIA analysis of TCGA database. (**C**) Binding sites between miR-9-3p and the 3′UTR of MDM2 predicted by TargetScan website. (**D**) RIP experiment finding that AGO2 could pull down MDM2. (**E**) Targeting relationship between miR-9-3p and MDM2 assessed by dual-luciferase reporter gene assay. (**F**) The expression of MDM2 in T24 cells after silencing both SNHG1 and miR-9-3p detected by qRT-PCR. (**G**) The expression of MDM2 in T24 cells after silencing both SNHG1 and miR-9-3p detected by Western blot analysis. (**H**) The expression of SNHG1, miR-9-3p and MDM2 after overexpressing MDM2 and silencing SNHG1 in T24 cells measured by qRT-PCR. (**I**) CCK-8 assay of the change of T24 cell viability. (**J**) The changes of T24 cell proliferation detected by EdU assay. (**K**) qRT-PCR detection of the expression of SNHG1 and miR-9-3p in T24 cells. (**L**) Tumor weight. * *p* < 0.05. N = 10 mice in each group. The cell experiment was repeated three times. Data were presented as mean ± standard deviation and compared by one-way analysis of variance or unpaired *t* test.

**Figure 8 cancers-14-04740-f008:**
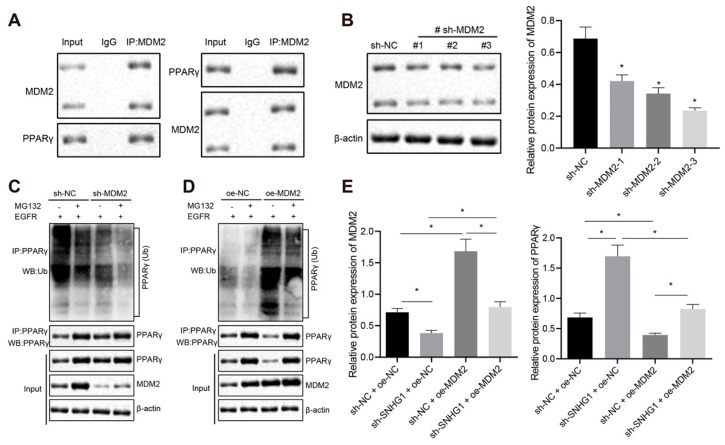
Silencing of SNHG1 decreases PPARγ ubiquitination and enhances PPARγ expression via MDM2. (**A**) The binding of MDM2 to PPARγ detected by IP assay in T24 cells. (**B**) Western blot analysis of the expression of MDM2 and screening of a shRNA sequence with the highest silencing efficiency. (**C**) IP assay detecting the change of PPARγ ubiquitination after silencing MDM2. (**D**) IP assay detecting the change of PPARγ ubiquitination after overexpressing MDM2. (**E**) MDM2 and PPARγ expression detected by Western blot analysis after overexpressing MDM2 and silencing SNHG1 in T24 cells. * *p* < 0.05. The cell experiment was repeated three times. Data were presented as mean ± standard deviation and compared by one-way analysis of variance.

**Figure 9 cancers-14-04740-f009:**
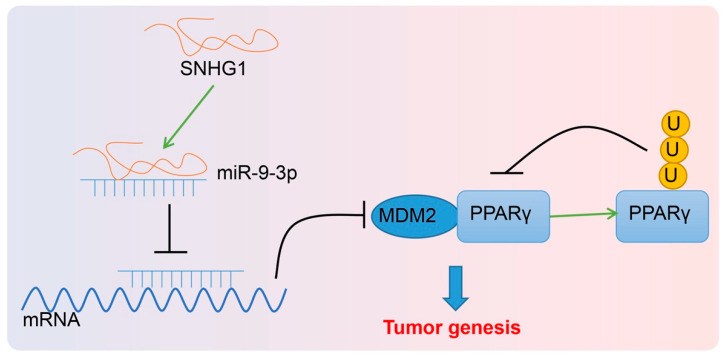
Flow map. SNHG1 promotes MDM2 expression by binding to miR-9-3p to promote PPARγ ubiquitination and downregulate PPARγ expression, thereby resulting in elevation of bladder cancer cell proliferation in vitro and tumorigenesis in vivo.

**Table 1 cancers-14-04740-t001:** Core degree of input genes in PPI network.

Gene	Degree	Gene	Degree
MDM2	28	SAMD4A	6
ZBTB10	14	EPHA7	5
KLF5	14	SRGN	4
SRP19	13	BNC2	4
ID4	9	RNF19A	4
CPEB4	9	MKX	3
CPEB3	9	CUTC	2
ETF1	9	DIXDC1	2
NTNG1	9	RIC3	1
INHBB	9	IGF2BP3	1
FBXL3	6		

**Table 2 cancers-14-04740-t002:** Primer sequences for qRT-PCR.

Targets	Forward Primer (5′-3′)	Reverse Primer (5′-3′)
SNHG1	GCCAGCACCTTCTCTCTAAAGC	GTCCTCCAAGACAGATTCCATTTT
miR-9-3p	GGAGACCGGAAATGTAGCCA	AATGGCCCGTGGAGTCTTTG
MDM2	GCAGTGAATCTACAGGGACGC	ATCCTGATCCAACCAATCACC
U6	CTCGCTTCGGCAGCACA	AACGCTTCACGAATTTGCGT
GAPDH	GGGAGCCAAAAGGGTCATCA	TGATGGCATGGACTGTGGTC

**Table 3 cancers-14-04740-t003:** sh-RNA sequences for transfection.

shRNAs	Sequences
sh-SNHG1#1	5′-GCTGAAGTTACAGGTCTGA-3′
sh-SNHG1#2	5′-GACCTAGCTTGTTGCCAAT-3′
sh-SNHG1#3	5′-GCTGAAGTTACAGGTCTGA-3′
sh-MDM2#1	5′-GCTTCGGAACAAGAGACTC-3′
sh-MDM2#2	5′-GGAGCAGGCAAATGTGCAATA-3′
sh-MDM2#3	5′-GGAACTTGGTAGTAGTCAATC-3′
sh-NC	5′-TTCTCCGAACGTGTCACGTTT-3′

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
