# Peer review of "LncRNA SNHG1 Facilitates Tumor Proliferation and Represses Apoptosis by Regulating PPARγ Ubiquitination in Bladder Cancer"

_cancers, 2022, doi:10.3390/cancers14194740_

Round 1

Reviewer 1 Report

The manuscript by Cai and co-authors examines the role of lncRNA SNHG1 on bladder tumor cell proliferation. This is a very good study which methodically delineates modulation of this lncRNA to changes in miRNA-9-3p, then MDM2 and finally to changes of PPAR gamma levels due to MDM2-dependent ubiquitination. The data is convincing and beautifully presented and warrants publication.

There are a few details that need to be addressed.

In the introduction the authors state that the reason for sex disparity where men are about 3 x more likely to develop disease is due to environmental factors. While environmental factors have a role, hormonal differences, particularly androgens, effect cancer development. This is reviewed in several publications and should be mentioned.

Figure 7B is unreadable. When amplified, the writing is too blurry. Please modify.

Figure 7G results show the quantitation of protein levels. Please include the western from which the quantifications were derived.

Bcl-2 and Bax are inversely regulated in several experiments. Can the authors provide some insight into the potential mechanisms driving these changes in the discussion?

Please review the language and grammar since there a few mistakes.

Author Response

Response to Reviewer 1 Comments

There are a few details that need to be addressed.

Point 1: In the introduction the authors state that the reason for sex disparity where men are about 3 x more likely to develop disease is due to environmental factors. While environmental factors have a role, hormonal differences, particularly androgens, effect cancer development. This is reviewed in several publications and should be mentioned.

Response 1: Thanks for your suggestion. We have modified this part according to your suggestion. “The increased risk for bladder cancer correlates to factors including age, smoking, exposure to some industrial chemicals and hormonal differences, particularly androgens” (Elizabeth MR, Laura CB, Guadalupe GA, Jesica EC. (2021). The Role of Androgens and Androgen Receptor in Human Bladder Cancer. Biomolecules. 2021 Apr 18;11(4):594.)

Point 2: Figure 7B is unreadable. When amplified, the writing is too blurry. Please modify.

Response 2: Thanks for your suggestion. We uploaded clearer pictures as additional materials for submission.

Point 3: Figure 7G results show the quantitation of protein levels. Please include the western from which the quantifications were derived.

Response 3: Thanks for your suggestion. Fig (G) The expression of MDM2 in T24 cells after silencing both SNHG1 and miR-9-3p detected by western blot analysis. And we have uploaded this part of data to the magazine as original data.

Point 4: Bcl-2 and Bax are inversely regulated in several experiments. Can the authors provide some insight into the potential mechanisms driving these changes in the discussion?

Response 4: Thanks for your advice. Our experiment is only a preliminary exploration of the relationship between SNHG1 and bladder cancer. For Bcl-2 and Bax, further experiments are needed to verify them. Our future work is one of them.

Point 5: Please review the language and grammar since there a few mistakes.

Response 5: Thanks for your suggestion. We have corrected some errors in the article.

Reviewer 2 Report

In general,

1. Μethodology part needs improvement. Many elements are omitted and some are only mentioned in the results creating confusion.

2. Τhe results are not completely understood. 

3. Be careful with the figures. They are very dense and difficult to understand. I suggest to simplify and or extend them. 

4. See in attached file more detailed comments

Author Response

Response to Reviewer 2 Comments

Introduction

Point 1: Globocan 2012 needs an uplate, there are data based on Globocan 2020 and bladder cancer -more recent data need to be presented

Response 1: Thanks for your suggestion. We have updated the data and revise our paper manuscript as “Bladder cancer (BCa) ranks 12h in cancer incidence with nearly 570000 new cases each year and 13th in terms of deaths ranks across the world” (Global Cancer Statistics 2020: GLOBOCAN Estimates of Incidence and Mortality Worldwide for 36 Cancers in 185 Countries)

Point 2: Lines 51-52: ‘observed the suppressive effect of SNHG1 on prostate cancer development by promoting cell proliferation’? à seems opposite, please check again (reference number 8)

Response 2: Thanks for your attention. We have checked again, SNHG1 negatively regulates miR-199a-3p to enhance CDK7 expression and promote cell proliferation in prostate cancer. It is correct.

Point 3: Lines 53, 54: It is not clear how you reach to this conclusion: “This evidence indirectly supported that SNHG1 might promote development of bladder cancer”.  The abovementioned studies describe pancreatic and prostate cancers.

Response 3: Thank you for your question. Our view is inferred from other literature reports, and it is “might” we used to express a possibility, but not certainty.

Methods

Point 1: 2.1.: Numbers and year of Ethics Committee approvals for both human and animal samples used should be written.

Response 1: Thanks for your suggestion. We have changed the paragraph as “The experiments involving 67 human beings from 2016 to 2017 were approved with ratification of Ethics Committee of Nanjing Medical University by conforming to the principles outlined in the Declaration of Helsinki. Ethical agreements were obtained from the donors or their families through written informed consent. Animal experiments (10 healthy nude mice aged 6-8 weeks) were ratified by Animal Ethics Committee of Nanjing Medical University and concurred with the Guidelines for Animal Experiments of Peking University Health Science Center.”

Point 2: 2.2.: 67 patients are included. Clinicopathological parameters about tumor, size grade, stage and age are not included. Follow-up intervals are also not included.

Response 2: Thanks for your suggestion. The relevant data are shown in Figure 1F. Our experiment mainly focused on the molecular mechanism of SNHG1 in bladder cancer. The clinical data analysis of 67 cases showed that SNHG1 was significantly overexpressed in bladder cancer and correlated with poor prognosis.

Point 3: Although you mention 67 patients, you refer to 40 patients only. Please explain what about 27 additional patients.

Response 3: Thank you for your question. Our date as showed in the Figure 1F, there were 67 cases (33 SNHG1 high expression and 34 SNHG1 low expression).

Point 4: 2.6. Cell transfection text: Table 1? First table is Table 2 with primer sequences for qRT-PCR.

Response 4: Thank you for your question. We have changed in numerical order in the manuscript.

Point 5: Tables 2, 3?

Response 5: Thank you for your question. We have changed in numerical order in the manuscript.

Point 6: 2.7.: normalization to GAPDH or U6 expression. Why one or another and when, please specify and explain the reasons.

Response 6: Thanks for your question. After recording of the Ct value of each well, the relative expression of mRNAs or miR was calculated using the 2-ΔΔCt method by normalizing to glyceraldehyde-3-phosphate dehydrogenase (GAPDH) or U6 expression.

Point 7: Transfection only of T24, 5367 cell lines. The other cell lines were not transfected? Based on what criteria T24 and 5367 cell lines were selected?

Response 7: Thanks for your question. We have selected 5637, T24, SW780 and UM-UC-3. (Line 136). We selected T24 and 5367 cell lines based on the SNHG1 expression.

Point 8: 2.9.: number of positive cells? Please clarify what do you mean by positive cells. It has to be clear.

Response 8: Thanks for your suggestion. The positive cells mean that the survived cells after sealing by the method of EdU.

Point 9: 2.10.: Which types of cells were evaluated by flow cytometry, please specify!

Response 9: Thanks for your suggestion. The T24 and 5367 cell lines were evaluated by flow cytometry and we added them in the Line 197.

Point 10: 2.12.: It is not clear which cells or cell lines were transfected with WT- or MUT MiR-9-3p. Please it must be clarified.

Response 10: Thanks for your suggestion. The T24 and 5367 cell lines were transfected with WT- or MUT MiR-9-3p.

Point 11: 2.13. Better writing. Specify cell lines used in this procedure, qRT-PCR normalization was performed with which gene, as well as the term “NC” must be explained!

Response 11: Thanks for your suggestion. T24 cell lines used in the procedure, and we have revised this part of the article according to your request.

Point 12: 2.14.: HEK293T cells are mentioned. However, there is no other information about. Please specify! The other cell lines were not used? If so, explain why!

Response 12: Thanks for your suggestion. HEK293 cells rarely express endogenous receptors required for extracellular ligands, and are relatively easy to transfect. It is a very common cell line for the expression of foreign genes. So it is often used as a tool cell. It is used in Dual Luciferase Reporter Gene Assay.

Point 13: 2.15.: IP assay: You write in line 232: cells…(What kind of cells, it is not clear). Also, in line 235: “the same amount of protein was taken from each experimental group”. Which experimental groups, it is not clear!!!

Response 13: Thanks for your question. The T24 cell lines were evaluated in our experiments. And we revised some parts of the article according to your request.

Point 14: 2.16. Subcutaneous Tumorigenesis Model in Nude Mice: line 249: ‘Approximately 2 × 106 cells were suspended in 200 μL PBS, and then subcu-249 taneously injected into the left or right hindlimbs of nude mice (10 mice/group)”. Again, the procedure is not clear, what kind of cells, which cell line?

Response 14: Thanks for your question. The T24 cell lines were evaluated in our experiments. And we revised some parts of the article according to your request.

Point 15: Figure 1 d, e: 40 patients from 67 were studied. They are the same 40 patients in all experiments? Please provide clinicopathological data of patients.

Response 15: Thanks for your question. As showed in Fig 1F, the clinical data analysis of 67 cases showed that SNHG1 was significantly overexpressed in bladder cancer and correlated with poor prognosis. Fig 1 D,E is the expression of SNHG1 in patients.

Results

When you write the term “expression” it must be clear if it is mRNA or protein expression. Please clarify in the text.

Point 1: Figure 2: T24 cell line,

2B: further explanation needed about 1-1, 1-2, 1-3 sh-SNHG1 differences and their selection.

2D: further explanation needed.

Also, in Figure 2 it is mentioned: **. However, there is nowhere **, so it must be 

removed.

Response1:Thanks for your suggestion. The differences and their selection between sh-SNHG1( 1-1, 1-2, 1-3)have been shown in table 1.** has been deleted.

Point 2: Figure 3: 5637 cell line

Based on which criteria these two cell lines were selected? 

Do you have an explanation about the differences on SNHG1 expression levels that they present? Is there a connection to their clinicopathological characteristics?

Also, in Figure 3 it is mentioned: **. However, there is nowhere **, so it must be removed.

Response 2:Thanks for your suggestion. To further examine the regulatory role of SNHG1 in bladder cancer, we selected one normal urothelial cell line SV-HUC-1 and four bladder cancer cell lines (5637, T24, SW780, and UM-UC-3). As described in Fig. 2A, SNHG1 expression was increased in cancer cells compared with SV-HUC-1 cells. Subsequent experiments were conducted on T24 cells with higher SNHG1 expression and 5637 cells with lower SNHG1 expression.** has been deleted.

Point 3: Figure 3: Better writing, explanation of terms such as oe-, NC-… in the image captions.

Response 3 :Thanks for your suggestion. Relevant notes have been marked.

Point 4: Figure 4: it is mentioned that 5637 cell line was used. However, this information isn’t mentioned in 2.16 of methods!

Response 4:Thanks for your suggestion. 2.16 shows the methods of tumorigenesis model in nude mice.

Point 5 : Figure 5: “(E) qRT-PCR detection of the expression of miR-9-3p after silencing SNHG1 in T24 cells. (F) qRT-PCR detection of the expression of miR-9-3p after overexpressing SNHG1 in 5637 cells”.

Why have you used 2 different cell lines (T24, 5637) for SNHG1 silencing or overexpression respectively? Please explain!

Figure 5: Better explanation in the caption.

Response 5:Thanks for your suggestion. As described in Fig. 2A, SNHG1 expression was increased in cancer cells compared with SV-HUC-1 cells. Subsequent experiments were conducted on T24 cells with higher SNHG1 expression and 5637 cells with lower SNHG1 expression. So we used 2 different cell lines (T24, 5637) for SNHG1 silencing or overexpression respectively.

Point 6: Line 381: it is not mentioned before miR-9-3p inhibitor. Please proceed to corrections in methods.

Response 6 :Thanks for your advice. Relevant methodologies are updated in methods.

Point 7 :Figure 6: miR-9-3p inhibitor co-treatment: how and where, in which cell lines, please clarify!

6B: not clear, further explanation needed!

6F,G: “antagomir” not mentioned before!

Response 7:Thanks for your suggestion. miR-9-3p inhibitor was used in T24 cells.Co-treatment with sh-SNHG1 and miR-9-3p inhibitor reversed the effect of sh-SNHG1 or miR-9-3p inhibitor alone in cck-8 assay. Antagomir Has been revised in the text line 412.

Point 8 : In all animal experiments the number of mice that were used is ten? Please, specify!

Response 8:Thanks for your patience. All animal experiments the number of mice that were used is 20.

Point 9 : Line 395: “the cell experiment..” please specify!

Response 9:Thanks for your patience. Related has been revised in the text.

Point 10: Figure 7b/ table 1: do not match,

Response 10:Thanks for your patience. Related has been revised in the text line 412.

Point 11:Tables are not in the right position in the text!!!

Response 11:Thanks for your patience. Related has been revised in the text.

Point 12: Figure 7: A, B: not clear pictures, please see resolution!

Response 12:Thanks for your patience. Related has been revised in the text.

Point 13: In Figure 7: Lanes 418-419 not in bold as in the other figures!

Response 13:Thanks for your patience. Related has been revised in the text.

Point 14: See tables 1 and 3, are not correct!

Response 14:Thanks for your patience. Related has been revised in the text.

Point 15: Figure 7D: What do you mean by fold enrichment? Where is MDM2 presented? 

Response 15:Thanks for your patience.AGO2 pulled down MDM2 in RIP experiments, fold enrichment means MDM2 pulled down by AGO2. MDM2 enrichment in nucleus.

Point 16: Line 431: L (tumor weight) needs parentheses ()!

Response 16:Thanks for your patience. Related has been revised in the text.

Point 17: Please check lines 436, 438-439 and 441, they seem to have opposite meaning!

Response 17: Thanks for your patience. These contents are picture annotations, which do not specifically describe the data correlation between groups, so there is no inconsistency in the results.

Point 18: Line 452: “As presented in Figure… X???

Response 18: Thanks for your patience. We checked the sentences and confirmed that there was no obvious problem with the relevance of the expressions.

Point 19: Figure 8A: What are the differences? Further explanation needed

Response 19: Thanks for your patience. Figure 8. Silencing of SNHG1 decreases PPARγ ubiquitination and enhances PPARγ expression via MDM2. And Figure 8A:The binding of MDM2 to PPARγ detected by IP assay in T24 cells.

Round 2

Author Response

Response to Reviewer Comments

Dear reviewer:

We appreciate your further revision of our article. Many details have been well revised, making our manuscript more academic and accurate. Thank you for your precious time, thank you very much, and wish you well.

Methods

Point 1. 2.1.: The protocol numbers/year of Ethics Committee approvals for both human and animals need to be written.

Response 1: Thanks for your suggestion. We added the protocol number/year of Ethics Committee approvals for both human and animals in part of Methods 2.1 as follow: “The experiments involving 67 human being samples were approved with ratification of Ethics Committee of Nanjing Medical University (NO. 402, 2021) by conforming to the principles outlined in the Declaration of Helsinki. Ethical agreements were obtained from the donors or their families through written informed consent. Animal experiments (20 healthy nude mice aged 6-8 weeks) were ratified by Animal Ethics Committee of Nanjing Medical University (2021) and concurred with the Guidelines for Animal Experiments of Peking University Health Science Center. “. And the ethical approval document has been uploaded to the magazine at the time of submission.

Point 2. 2.7.: Normalization to GAPDH or U6 expression: Specify in the text when did you use

GAPDH or U6 as a reference for normalization.

Response 2: Thanks for your advice. The part of 2.7 we did not pay attention to distinguish the use of GAPDH and U6. Thank you for your attention. We have modified it in the paragraph.

Point 3. Transfection only of T24, 5367 cell lines. The other cell lines were not transfected.  Based on what criteria T24 and 5367 cell lines were selected? RESPONSE 7 should be added in the text!!!

Response 3: Thanks for your advice. We added the explanation of the criteria T24 and 5367 cell lines selected in the line 177 as follow: “the expression of SNHG1 was obvious highest in T24 and compared with sw780 cells, 5637 cells had better culture characteristics in our experiment”

Point 4. 2.15.: IP assay: cells…(What kind of cells, it is not clear). Also, in line 235: “the same amount of protein was taken from each experimental group”. T24 cells should be mentioned clearly in the text!

Response 4: Thanks for your advice. We have revised the article as follow:” T24 cells were lysed in lysis buffer [mixture of 50 mM Tris-HCl (pH 7.4), 150 mM NaCl, 10% glycerol, 1 mM ethylene diamine tetraacetic acid, 0.5% NP-40, and protease inhibitor], and cell debris was cleared by centrifugation. After the concentration of lysis was measured by BCA, the same amount of protein was taken from oe-NC/OE-MDM2 group and replenished to the same volume with cell lysate.”

Point 5. 2.16. Subcutaneous Tumorigenesis Model in Nude Mice: ‘Approximately 2 × 106 cells were suspended in 200 μL PBS, and then subcutaneously injected into the left or right hindlimbs of nude mice (10 mice/group)”. Again, the procedure is not clear, what kind of cells, which cell line? T24 cell line should be written clearly in the text!

Response 5: Thanks for your advice. It was 5637 cells and we added it in line 261.

Results

Point 1. Figure 4: it is mentioned that 5637 cell line was used. However, this information isn’t mentioned in 2.16 of methods! Please add at section 2.16 that 5637-cell line was used!!!

Response 1: Thanks for your advice. We have added it in the section.

Point 2. Figure 6: “The miR-9-3p inhibitor was used in T24 cells. Co-treatment with sh-SNHG1 and miR-9-3p inhibitor reversed the effect of sh-SNHG1 or miR-9-3p inhibitor alone in cck-8 assay.” Add these lines in the text, in order to be clear!!!

Also, the term antagomir should be clearly written at materials/methods!!!

Response 2: Thanks for your request. We have added these lines in the text (line 395-396), and we added the antagomir in the materials/methods (line 262)

  1. In Figure 7: SNHG1 silencing decreases MDM2 expression via miR-9-3p to repress bladder cancer cell proliferation and tumorigenesis. Make bold as in the other figures!

Response 3: Thanks for your attention. It is our mistake, we have made bold.  

  1. Figure 8A: The binding of MDM2 to PPARγ detected by IP assay in T24 cells. Explain

in the text what represent clearly the two figures at 8A!!!

Response 4: Thanks for your advice. We have revised this paragraph as follow:” Firstly, through IP experiments in T24 cells, it was found that MDM2 combined with PPARγ. Meanwhile, the combination of PPARγ to MDM2 can also be proved(Fig. 8A).”
